

# Hypergravity hinders axonal development of motor neurons in *Caenorhabditis elegans*

Saraswathi Subbammal Kalichamy, Tong Young Lee, Kyoung-hye Yoon and Jin Il Lee

Division of Biological Science and Technology, Yonsei University, Wonju, South Korea

## ABSTRACT

As space flight becomes more accessible in the future, humans will be exposed to gravity conditions other than our 1G environment on Earth. Our bodies and physiology, however, are adapted for life at 1G gravity. Altering gravity can have profound effects on the body, particularly the development of muscles, but the reasons and biology behind gravity's effect are not fully known. We asked whether increasing gravity had effects on the development of motor neurons that innervate and control muscle, a relatively unexplored area of gravity biology. Using the nematode model organism *Caenorhabditis elegans*, we examined changes in response to hypergravity in the development of the 19 GABAergic DD/VD motor neurons that innervate body muscle. We found that a high gravity force above 10G significantly increases the number of animals with defects in the development of axonal projections from the DD/VD neurons. We showed that a critical period of hypergravity exposure during the embryonic/early larval stage was sufficient to induce defects. While characterizing the nature of the axonal defects, we found that in normal 1G gravity conditions, DD/VD axonal defects occasionally occurred, with the majority of defects occurring on the dorsal side of the animal and in the mid-body region, and a significantly higher rate of error in the 13 VD axons than the 6 DD axons. Hypergravity exposure increased the rate of DD/VD axonal defects, but did not change the distribution or the characteristics of the defects. Our study demonstrates that altering gravity can impact motor neuron development.

# INTRODUCTION

Human anatomy and physiology is well adapted for survival and fitness on the planet Earth and its stable gravitational conditions. Altering gravity can have profound impacts on the human body. This is especially relevant with the possibility of long-term space travel and habitation and the associated changes in gravity in different space environments. Although some of the effects of decreased gravity on the human body are identified, such as muscle atrophy (*Vandenburgh et al., 1999*), the genetic, molecular and cellular basis of gravity-induced changes is not well known.

One of the cellular processes that may be constrained by gravity conditions is neuronal development. Previous work showed that microgravity induced a decrease in synaptic

Corresponding author
Jin Il Lee, jinillee@yonsei.ac.kr

density in the hindlimb area of the motor cortex in the brains of rats that were aboard a space mission for 16 days (*DeFelipe et al., 2002*). Alterations in gravity could also affect the development of neuronal projections. In hypergravity conditions, 150G force induced neuron-like cell differentiation and development of longer neurites in cultured PC12 cells (*Genchi et al., 2015*). Moreover, low gravity during spaceflight can induce alterations in dendritic arbor development in medial spinal motor neurons in rats that innervate and control muscle (*Inglis, Zuckerman & Kalb, 2000*). Although altering gravity can affect motor neuron development, the genetic and molecular mechanism of gravity's effect on neuron development is unknown.

The nematode *C. elegans* is one of the premier metazoan genetic model organisms, and studies with the worm have led to seminal discoveries including RNA interference, microRNAs, and genes controlling programmed cell death (*Fire et al., 1998*; *Hengartner & Horvitz, 1994*; *Lee, Feinbaum & Ambros, 1993*). Moreover, because worms are easy to handle, studies using *C. elegans* have been conducted in space. Worms survive and grow well in space, and many functions of the worms are normal (*Higashitani et al., 2005*; *Szewczyk et al., 2008*; *Zhao et al., 2006*). Muscle gene expression is slightly altered in space (*Higashibata et al., 2006*), as well as markers for aging (*Honda et al., 2012*). Hypergravity experiments on the ground show that behaviors and muscle structure of the worms are normal at a 100G force, although the FOXO transcription factor DAF-16 translocates to the nucleus (*Kim et al., 2007a*). However, no thorough studies on the effects of altering gravity to the motor neurons have been conducted.

Unlike vertebrate motor neurons that can exhibit plasticity (*Inglis, Zuckerman & Kalb, 2000*), *C. elegans* motor neurons are stable and fixed during adulthood (*Sulston, 1976*; *White et al., 1976*). Motor neurons innervate four blocks of body wall muscle that line the ventral, dorsal, and lateral sides of the worm body to control forward and backward movement (*Sulston & Horvitz, 1977*). Since most of the changes in the motor neurons occur during development, the motor system in developing *C. elegans* is a better model for changes that occur in the adult vertebrate motor neurons than adult *C. elegans*. Particularly, D-type GABAergic motor neurons that are found on the ventral side of the animal extend circumferential axonal processes called commissures to the dorsal side where they join with the dorsal nerve cord to control movement (*White et al., 1976*). Six D-type motor neurons called the DD neurons are born embryonically and extend commisssures dorsally before the worm hatches from the egg shell. Another 13 D-type neurons called the VDs are born late in the first larval stage and then extend their commissures dorsally (*White et al., 1976*). Axon growth cones from the developing DD/VD motor neurons sense secreted attractive or repulsive cues along the body wall to find their targets in the dorsal side of the worm (*Colavita et al., 1998*; *Hedgecock, Culotti & Hall, 1990*; *MacNeil et al., 2009*).

The proper development of all 19 DD/VD motor neurons requires many genes expressed from multiple cell types, and can be easily altered by genetic manipulation (*Wadsworth, 2002*). We wondered what the effect of altering gravity would have on the development of the DD/VD neurons. In this study we increased the gravity force on developing *C. elegans* worms by centrifugation, resulting in a hypergravity environment.

We found high gravity significantly increases the number of animals that had disrupted DD/VD axon projections, and we characterized the defects we found in detail here.

## MATERIALS AND METHODS

### Nematode culture and strains

Animals were grown and maintained at 20 °C on Nematode Growth Medium (NGM) plates seeded with *E.coli* OP50 as described previously (*Brenner, 1974*). Strains used for this study: N2, LG II: *juIs76* ((p)*unc-25*::GFP), LG IV: *evIs82a* ((p)*unc-129*::GFP) and LG X: *zdIs5* ((p)*mec-4*::GFP). The genetic background of the *juIs76* strain was maintained by backcrossing this strain with the N2 wild-type strain, which was effective in maintaining a consistent axonal defect rate (see section Quantification of Defects).

### Nematode hypergravity cultivation tube

A total of 1 L NGM media was prepared similar to previous protocols (*Lewis & Fleming, 1995*), except Difco granulated agar was substituted for Bactoagar (*Lee, Yoon & Lee, 2016*). The 1 ml of NGM was placed into each 1.5 ml centrifuge tube and then transferred to a heat block set at 42 °C to prevent immediate solidification of agar. After distribution into tubes, the tubes were spun at 100G for 2 min to solidify the agar and create a surface in which the force of gravity is perpendicular to the flat agar surface in the tube.

To create a small lawn of *E. coli* bacteria, a single colony of OP50 strain bacteria was inoculated in LB broth and incubated in a shaker at 37 °C overnight, then concentrated by spinning down and removing the supernatant. The pellet was resuspended and 2 μl of bacteria was added to the surface of each tube and allowed to dry at room temperature for at least 24 h before usage.

### Preparation of eggs and hypergravity exposure

Eggs were harvested by bleaching gravid worms according to a standard protocol (*Steiernagle, 1999*), and then washed and collected in M9 buffer. The 1 μl of this solution was counted to obtain an approximate egg population density. For hypergravity experiments, 100–150 eggs were placed into the cultivation tube, and centrifuged in a temperature-controlled micro-centrifuge (Sorvall Legend Micro17R Centrifuge; Thermo Scientific) at the particular G-force (rpm values for 6, 10, 100 and 500G are 300, 400, 1,200 and 2,800 rpm, respectively) at 20 °C for various exposure times. A control tube (1G) maintained in a 20 °C incubator was performed with every experiment. Worms were assessed by microscopy at the L4 larval/young adult stage which is 60 h after egg harvesting. If the hypergravity exposure time was shorter than 60 h, the tubes were placed in a 20 °C incubator after the hypergravity exposure until 60 h was reached. For 3 h acute exposure experiments, three-day-old adult animals were exposed to 100G hypergravity for 3 h. For 60 h exposure adult animal experiments, individual six-day-old adult animals were picked directly into the cultivation tube and exposed to 100G hypergravity for 60 h.

### Microscopic analysis

After 60 h, animals were washed with M9 buffer and collected into 1.5 ml tubes and allowed to settle on the bottom of the tube. After removing the supernatant, animals were

mounted onto glass slides with a dry 2% agarose pad. To immobilize the worms, 2 μl of 1 M sodium azide was added onto the agarose pad. An epi-fluorescent microscope (Olympus BX50) was used to visualize the GFP-labelled neurons. Imaging software (Nikon Elements) was used for extended depth of field (EDF) images, as well as all others images.

## Quantification of defects

Defects were scored by researcher microscopic observations. Animals were scored as defective if one or more of the 19 DD/VD neurons showed any axonal defect (branch, turn and extend, stop, join and reach). Occasionally, an abnormally high axonal defect rate (50% or more) was observed even in normal 1G conditions. We censured and discarded all the data from these days. We found that this abnormality may be due to random genetic abnormalities within this strain's genome. Thus, we maintained the genetic background of the *juIs76* strain by backcrossing this strain with the N2 wild-type strain. Dorsal-ventral locations of defects were approximated by eye and scored as: Ventral—0 to 15% dorsal-ventral distance, Ventral sublateral—15% to 50% dorsal-ventral distance, Dorsal sublateral—50% to 85% dorsal-ventral distance, Dorsal—85% to 100% dorsal-ventral distance. Student's T-test was performed (Microsoft Excel) to analyse the significance of data. For statistical analysis with multiple sets of data, single-factor ANOVA was performed (Microsoft Excel) to determine variance, and post-hoc Bonferroni correction was performed to determine significance.

# RESULTS

## Hypergravity induces DD/VD motor neuron axonal defects

To study the effect of hypergravity on *C. elegans* biology, we used a tabletop refrigerated centrifuge to create a gravity force and designed a small worm cultivation tube from a 1.5 cm centrifuge tube filled with NGM agar and seeded with OP50 *E. coli* bacteria on top (Fig. 1A). To visualize the DD/VD motor neurons, we used a *C. elegans* transgenic strain that expresses GFP under the control of the *unc-25* gene promoter. The *unc-25* promoter directs expression of glutamic acid decarboxylase, an enzyme necessary for the production of GABA neurotransmitter, in the DD/VD motor neurons as well as several other neurons (*Jin et al., 1999*; *McIntire et al., 1993*). Using the (p)*unc-25*::GFP strain, a total of 19 commissural DD/VD axons can be observed running circumferentially from the ventral to the dorsal side (Fig. 1B). Eggs containing developing embryos were placed in the cultivation tube, and either spun in the centrifuge to induce a high gravity force of 100G or placed in a 20 °C incubator as a 1G gravity control. There were no obvious differences observed by eye between 1G and 100G: the pace of development in 1G and 100G was identical, and hermaphrodite worms at 100G seemed to develop normally to gravid adult mothers with no noticeable defects in movement. This is consistent with a previous study that cultivated *C. elegans* in a microfluidic compact-disc cultivation system at 100G and showed that worms had normal growth and adult movement and behaviors (*Kim et al., 2007a*; *Kim et al., 2007b*).

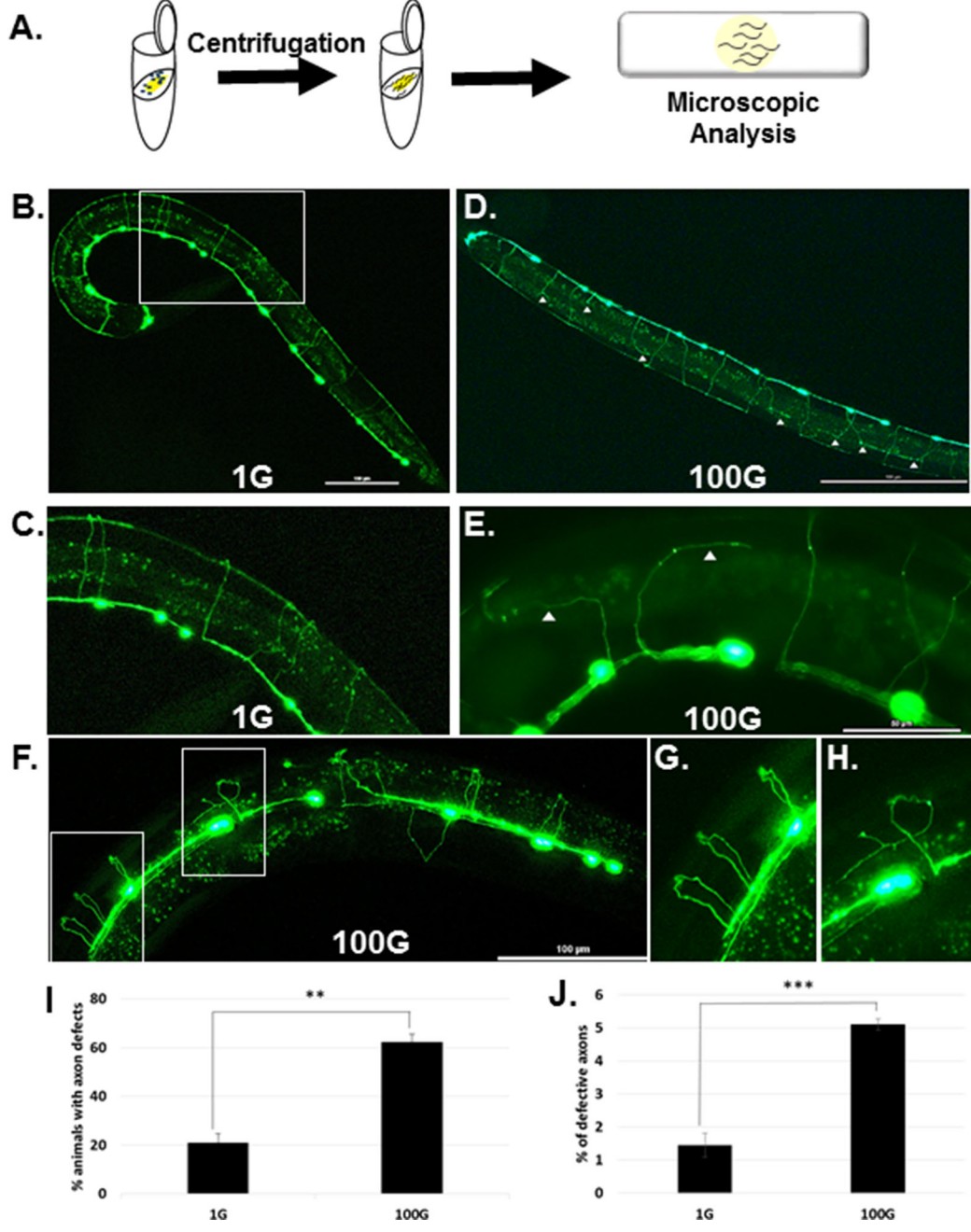

**Figure 1 In 100G, hypergravity induces axonal defects in DD/VD motor neurons.** (A) Harvested embryos were exposed to hypergravity by centrifugation and then analyzed by microscopy as adults for neuronal defects. (B) (p)*unc-25*::GFP control animal at 1G gravity shows normal axonal commissures. Bar = 100 μm (C) Magnified view of boxed area in (B). (D) (p)*unc-25*::GFP animal exposed to 100G hypergravity shows multiple axon commissural defects (white triangles) Bar = 100 μm. (E) Axonal defects in 100G exposed animals. Bar = 50 μm. (F) 3D extended depth of field (EDF) image of (p)*unc-25*::GFP exposed to 100G hypergravity. Ventral in front, dorsal in back. Bar = 100 μm. (G) Magnified image of white box in (F) showing normal axonal commissures. Note the circumferential axons traveling along the body wall from the ventral to dorsal side during development. (H) Magnified image of white box in (F) showing defective axon that turned and extended and formed branches before it approached the dorsal side of the animal. (I) Percent animals that display at least one axon defect for 1G and 100G. Error bars represent SE. T-test was performed and ** indicates statistical significance p < 0.005. (J) Percent axons that are defective in 1G and 100G. T-test was performed and *** indicates statistical significance p < 0.0005. Error bars represent SE.

After 60 h, the worms had reached mid-late L4 larval stage. At 1G, most of the animals showed normal DD/VD commissural projections that reached the dorsal nerve cord (Figs. 1B and 1C), although occasional defects could be observed (Figs. 1I and 1J). However, in animals exposed to 100G hypergravity for 60 h, defective axonal projections could be seen more frequently. When we quantified the defects, we found that 21% ± 3.75 (four trials, 425 total animals) of worms grown in 1G displayed axonal defects (Fig. 1I). However, worms grown in 100G had at least one axonal defect in over 60% ± 3.29 (four trials, 447 total animals) of the worms, which is about a 3-fold higher frequency than in 1G conditions (Fig. 1I). Among the total axons we looked at, axonal defects occurred at a frequency of 1.5% ± 0.36 (Fig. 1J; four trials, 8,046 total axons). In 100G, defects occurred at a rate of 5.1% ± 0.17 (four trials, 7,650 total axons), an over three-fold increase compared to 1G (Fig. 1J).

Next, we varied the hypergravity force to observe their effects on DD/VD axon development. Forces of 10 to 500G increased the percent of worms with axonal defects by approximately 30% over 1G. Interestingly, the number of defects did not significantly differ from 10 to 500G. On the other hand, a force of 6G, the minimum speed for our table top centrifuge, could not induce axonal defects (Fig. 2A). Therefore it seems that a certain threshold in gravitational force exists between 6G and 10G that induces axonal defects. For our experiments, we decided to use a hypergravity force of 100G, which is the force used in a previous study (Kim et al., 2007a).

## Hypergravity-induced axonal defects requires exposure during a specific period of development

We wondered if there was a critical period during development that hypergravity can induce motor neuron axonal defects. Hence, we exposed embryos to hypergravity for different time periods from the embryo stage and assessed DD/VD commissural axon defects in these animals at the 60 h point or 72 h point. Exposure of developing *C. elegans* to 100G hypergravity for 18, 60, and 72 h all resulted in an approximately 2.5–3 fold significant increase in axonal defects (Figs. 2B and 2C; n trials 18 h = 4, 60 h = 4, 72 h = 6). A shorter exposure of 10 h resulted only in a small increase in worms with axonal defects, and a 3 or 6 h exposure did not increase axonal defects (Figs. 2B and 2C; n trials 3 h = 5, 6 h = 5, 10 h = 4). We also exposed three-day-old adult *C. elegans* to 100G hypergravity for 3 h, and 6-day old adult animals to 100G for 60 h, but there were no significant increases in axonal defects in the adult animals (n trials 3 h = 5, 60 h = 5). It is important to note that embryos recovered by bleaching the mothers are not at a synchronous stage of development. Instead, these embryos are within an approximately 6 h range of development. This may explain why we observe an intermediate phenotype at 10 h hypergravity exposure (Fig. 2C). Taking this into consideration, with our methods a minimum of 18 h in hypergravity is required for an increase in axonal defects.

What may be occurring in the DD/VD motor neurons during this time in development? DD neurons are born during the embryonic stage on the ventral side of the animal and send projections to the dorsal side well before hatching (Sulston, 1976). The VD neurons, however, are not born until the late L1/early L2 stage, more than 20 h

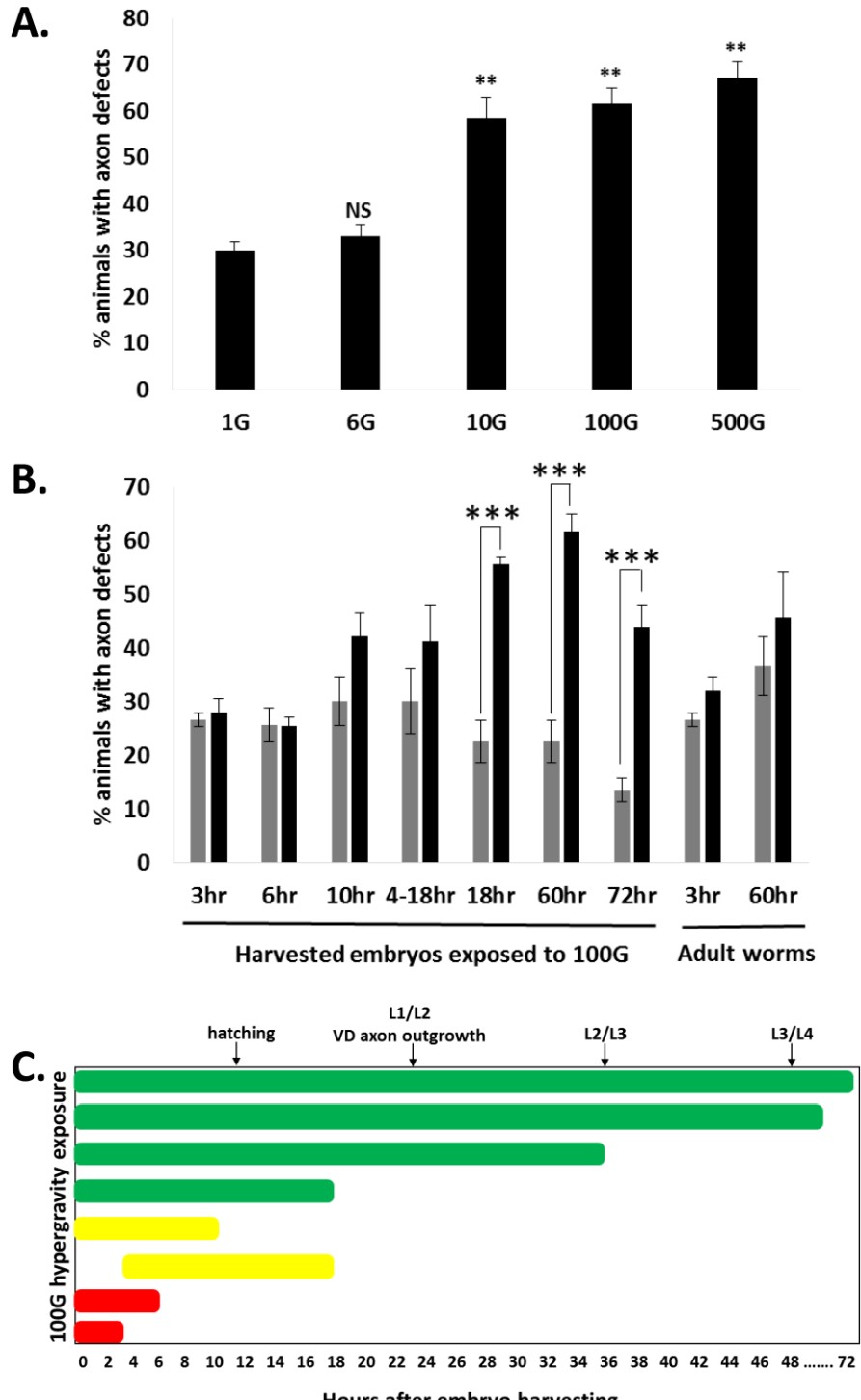

**Figure 2 Hypergravity force and exposure time affect DD/VD neuron axon development.** (A) Animals were exposed to 1–500G hypergravity for 60 h after embryo harvesting. Gravity force of over 10G increased axonal defects. T-test was performed and ** indicates statistical significance p < 0.0005. Error bars represent SE. (B) Animals were exposed to 100G hypergravity for various times after embryo harvesting, or to 3 h or 60 h during adulthood (far right bars). Grey bars = 1G control, black bars = 100G. Single-factor ANOVA and post-hoc Bonferroni correction was performed, ** indicates statistical significance p < 0.005. (C) Data in (B) represented by exposure time over the major developmental events of *C. elegans* (top of graph). Green represents axonal defects induced, red represents axonal defects not induced, yellow represents axonal defects slightly induced. Bars represent SE.

**Table 1 Axon defects in different neuron types at 1G and 100G.**

| Neuron type | Reporter strain | 1G | | 100G | |
|---|---|---|---|---|---|
| | | % defective axon | n | % defective axon | n |
| DD motor neurons | (p)*unc-25*::GFP | 0.98 | 630 | 2.16 | 631 |
| VD motor neurons | (p)*unc-25*::GFP | 2.75 | 630 | 6.21 | 631 |
| DD motor neurons (L1) | (p)*unc-25*::GFP | 0 | 375 | 0 | 316 |
| Mechanosensory neurons | (p)*mec-4*::GFP | 0 | 50 | 0 | 45 |
| Cholinergic neurons | (p)*unc-129*::GFP | 0 | 102 | 0 | 118 |

**Note:**
n = number of animals.

after these eggs are harvested from their mothers (Fig. 2C). Thus, during this 18 h critical exposure period, the VD neurons likely have not yet been born. More surprising was when we restricted exposure to 100G hypergravity from 4 h to 18 h, we saw a decrease in induced axonal defects compared to the full 18 h to a point (Figs. 2B and 2C; n trials = 4), further illustrating the importance of early exposure to hypergravity in the observed increase in axonal defects.

## Characterization of hypergravity-induced axonal defects

We then characterized the DD/VD motor neuron axonal defects more carefully by counting the defects in each commissure. We first checked the structure of the DD neurons in the L1 stage immediately after a 15 h hypergravity exposure and found that 100% of DD neuronal commissures are normal at this time point (Table 1). This was puzzling, since this meant that the defects we see in 1G at 60 h likely arise after the mid-L1 stage, either by some alteration of the DD axons, or some predisposed developmental change to the yet-born VD neurons. This led us to characterize each of the axonal defects in the DD/VD neurons induced by 60 h 100G hypergravity exposure more thoroughly.

There are a total of 19 commissures, DD1 to DD6 and VD1 to VD13, with VD1 being the most anterior commissure, and VD13 at the most posterior end (Fig. 3A). The DD1 and VD2 commissures overlay each other in the same commissural tract and are indistinguishable. Therefore, we considered DD1/VD2 as one commissure, and count a total of 18 DD/VD commissures. At normal 1G gravity conditions, occasional axon defects were observed in most commissures spanning the anterior and posterior ends of the worm (Fig. 3B). We found that certain commissures had a higher probably of having axon defects than others. For instance, the five commissures in the midbody region that include DD3, VD6, VD7, DD4, and VD8 accounted for 50.93% of the total axonal defects whereas the five commissures on the anterior or posterior end including VD1, VD2/DD1, VD12, DD6, and VD13 accounted for only 8.87% of the defects (n = 630 animals, total defects = 214). We also found that axonal defects were skewed towards the VD commissures and that DD neurons showed less defects. We observed that 1.9% of all the VD commissures were defective, whereas only 0.8% of the DD commissures showed defects (Table 1). In conclusion, although axonal outgrowth and guidance is a faithful biological process that allows neuronal projections to reach their targets, the development of the DD and particularly the VD motor neuron commissures are slightly error prone.

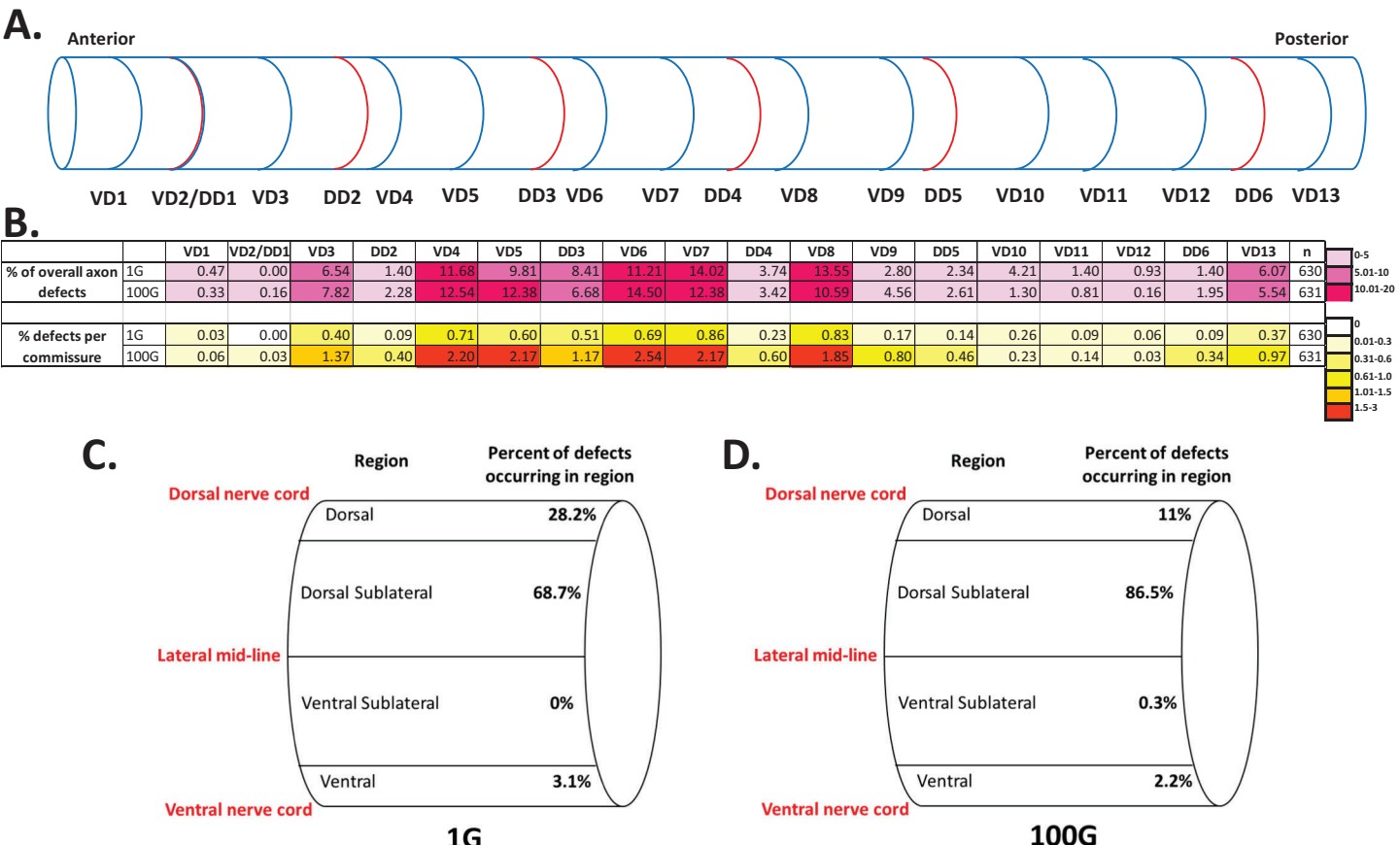

**Figure 3 Spatial distribution of axonal defects in 1G and 100G hypergravity exposed animals.** (A) Location of the DD and VD axon commissures along the anterior-posterior axis of the worm. VD2 and DD1 are overlapped commissures and are indistinguishable. (B) Axon defects for each VD/DD commissure. Top table shows the contribution of axon defects for each commissure to the total number of defects for 1G and 100G. The percents for each commissure add up to 100% for 1G and 100G, respectively. Heat maps in purple/lavender shades show the percents indicated in the legend on the right. N = number of worms. Bottom table shows the percent defects for each individual commissure for 1G and 100G. Heat maps in orange/yellow shades show the percents indicated in the legend on the right. (C) Localization of defects in the dorsal-ventral direction for 1G animals. Defects were categorized into the four regions listed and percent of defects in each region is shown. (D) Localization of defects in the dorsal-ventral direction for 100G hypergravity exposed animals.

While defects sporadically occur during the development of the DD and VD axons in 1G gravity conditions, 100G hypergravity clearly aggravates this error rate. In 100G, the rate of defective axons jumps from 0.98 to 2.75% for DD commissures and 2.16 to 6.21% for VD commissures (Table 1), a 2.81-fold and 2.88-fold increase, respectively. Although hypergravity induces more axonal defects, it does not significantly change the distribution of those errors. For every DD axonal defect there are about 2.2 VD defects at 1G, and this ratio is remarkably maintained in 100G. In respect to location along the worm body, the five midbody commissures still account for 58% of the axonal defects in 100G, whereas the five anterior/posterior end neurons again only account for 8% of the errors (n = 631 animals, total defects = 614). Hence, hypergravity only exacerbates the total axon defect rate.

Next, we determined the location of each of the axonal defects in the ventral/dorsal direction. We grouped the defects into four categories: Ventral, Ventral sub-lateral, Dorsal sub-lateral, and Dorsal. At 1G, we found that most of the defects (68.7%) were located in

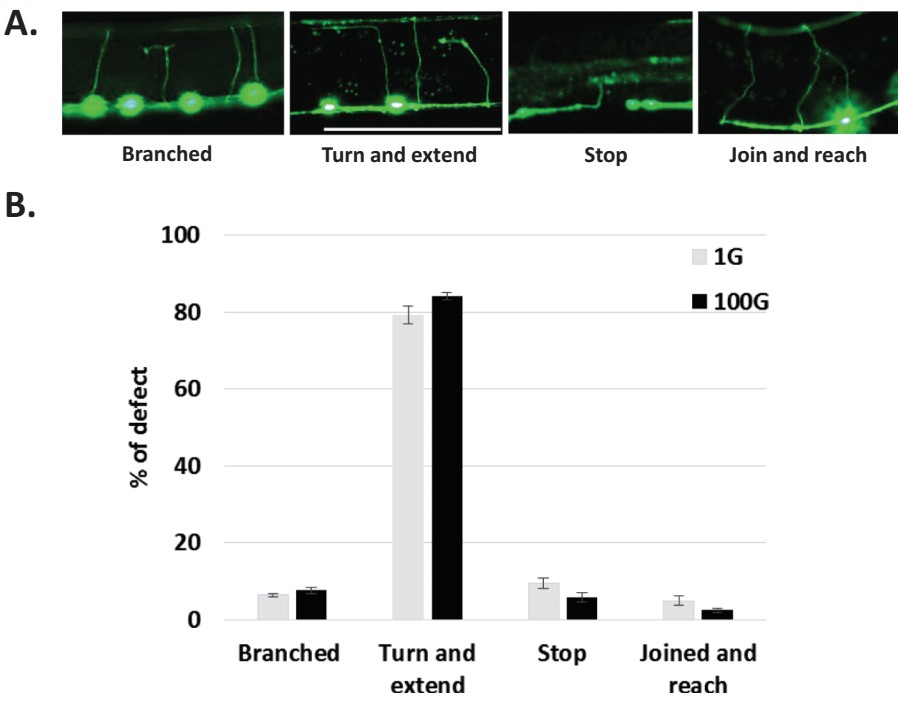

**Figure 4 Qualitative characterization of axonal defects in 1G and 100G hypergravity exposed animals.** (A) Axonal defects were categorized into the four groups shown here. (B) Quantification of axonal defects by category for 1G and 100G exposed animals. T-test was performed and none of the values were shown to be significantly different. Bars represent SE.

the dorsal sub-lateral region which is defined as the area dorsal of the lateral mid-line, but ventral to the area of the dorsal nerve cord (Fig. 3C, see section Materials and Methods). Interestingly, this dorsal sub-lateral bias in axonal defects was maintained or slightly higher at 100G (86.5%). We observed a decrease in the number of dorsal defects in 100G, however the overall dorsal side bias of errors (96.9% in 1G, 97.5% in 100G) is quite consistent. Once again, we find that hypergravity does not greatly alter the pattern of axonal defects observed in normal gravity conditions.

We also characterized the axon commissural defects based on morphology. We defined four categories of defects: branched, turn and extend, stop, and joined and reach (Fig. 4A). Axons that abruptly stop, branch, or turn and extend never reach the dorsal nerve cord, whereas axons that join together do reach the dorsal nerve cord. Most of the defect were in the turn and extend category at close to 80%, while others were all less than 10%. Interestingly, the distribution of axonal defects in 1G conditions was strikingly maintained in 100G hypergravity (Fig. 4B). Taken together, we find that hypergravity increases the overall rate of DD/VD commissural axon defects compared to normal gravity, but does not alter the distribution of axon defects.

## Hypergravity-induced axonal defects are specific for the DD/VD motor neurons

In a previous study, it was reported that 100G hypergravity did not affect the structure of the chemosensory neurons ASI and ADF, and touch sensory neurons AVM, ALM, PVM,

and PLM, as well as the function of the AWA olfactory sensory neurons (*Kim et al., 2007a*). We exposed a strain of *C. elegans* that expressed GFP in the touch sensory neurons to 100G and confirmed that the structure of the touch sensory neurons were not affected by 100G hypergravity in our cultivation system (Table 1). To determine whether the structure of other motor neurons could be affected, we exposed a strain of *C. elegans* that expressed GFP in the DA/DB cholinergic motor neurons. The axonal commissures from the DA/DB motor neurons project from the ventral to the dorsal side of the animal similar to the DD/VD motor neurons. However, 100G hypergravity for 60 h did not induce any defects in the DA/DB motor neuron axons (Table 1). Thus, the hypergravity-induced axonal defects we have observed and characterized may be specific for the DD/VD GABAergic motor neurons.

## DISCUSSION

The physiology and cellular functions of organisms are adapted to the 1G gravity conditions here on Earth. This is clearly demonstrated by the biological effects that altered gravity in spaceflight has on vertebrates, particularly muscle atrophy (*Vandenburgh et al., 1999*). However, broader effects of altered gravity at the cellular level remain obscure. Here, we evaluated the effect that high gravity has on motor neuron development in the nematode *C. elegans* and characterized these changes. From 10 to 500G hypergravity induces DD/VD motorneuron axon defects in 30% more animals than 1G. Most of the axon defects were found in the dorsal sublateral region, and none of the defective axons reached their targets in the dorsal nerve cord. We found that an 18 h exposure from the embryonic stage to the first larval stage was sufficient to cause the axonal defects, whereas an acute 3 h or longer 60 h exposure in the adult animal did not induce defects. Although hypergravity increases the overall DD/VD motor neuron axon defects compared to 1G, it does not alter the distribution or the characteristics of these defects. Finally, these defects seem to be specific for the DD/VD motor neurons.

Though we saw the effects of hypergravity at a range of G forces, we used 100G for most of our experiments. A previous study applied a 100G force using a compact disc-type cultivation apparatus (*Kim et al., 2007a*). This study showed that hypergravity induced the nuclear localization of the conserved FOXO transcription factor DAF-16, and this was dependent on the ENaC/degenerin sodium channel that functions in mechanosensation. Many features in the worm were preserved in 100G, including muscle structure, olfactory behavior, feeding behavior, and the structure of several neurons (*Kim et al., 2007a*) (Table 1). However, the effect of hypergravity on the structure of *C. elegans* motor neurons had not been assessed yet. Although our hypergravity cultivation system was different than the compact-disc type apparatus previously used (*Kim et al., 2007b*), we confirmed that growth progressed normally, movement was normal, and the structure of touch sensory neurons was normal (*Kim et al., 2007a*) (Table 1).

Although hypergravity can increase axonal defects in the VD motor neurons, we still observe a range of DD/VD axonal defects in a portion of the control animals at normal gravity. This is consistent with observations from other groups that have studied DD/VD commissures using the *juIs76* strain that expresses (p)*unc-25*::GFP (*Cáceres et al., 2012*;

*Lee et al., 2015*). In addition to the *juIs76* strain, we used the *oxIs12* strain that also expresses (p)*unc-47*::GFP in the DD/VD neurons (*McIntire et al., 1997*). Interestingly, we did not detect many DD/VD axonal defects at either 1G or 100G in this strain (see Supplemental Information). However, a previous report cautioned that the *oxIs12* transgenic strain interfered with dosage compensation and affected expression of X-linked genes involved in axon guidance such as *unc-6*/netrin and *lon-2*/glypican genes, altering axon guidance phenotypes in different genetic backgrounds (*Gysi et al., 2013*). In addition, previous studies using electron microscopy have shown that the DD motor neurons can develop in slightly different locations in individual animals (*White et al., 1976*), and the number of axons innervating the dorsal nerve cord can also vary in individual animals (*Hedgecock, Culotti & Hall, 1990*). These studies, along with the other studies using the *juIs76* strain, provide evidence that the development of D-type motor neurons may show slight differences between individuals. Thus, we have confidence that our observations using the (p)*unc-25*::GFP strain are valid, and this lends credence to the fact that DD/VD axon development may normally be error prone.

Two major questions remain unanswered: what causes the errors in DD/VD axonal development, and how is hypergravity increasing these errors? Clues to answer the first question may lie in the distribution of the axonal defects. We find that commissures in the mid-body region tend to show more defects than the commissures at the ends of the animal. One possibility that may cause this to arise is that the length of the commissures are slightly longer in the mid-body area than the ends of the animal allowing for the possibility for more errors. Other differences in the mid-body compared to the ends of the animal are structures such as gonads, vulva, and muscle that could interfere with the guidance or migration of the axons. In fact, a previous study showed that axonal regeneration is less robust in the mid-body region likely due to interfering structures (*Wu et al., 2007*). We also find more defects in VD axons compared to DD axons. DD axons develop much earlier than the VD axons, and we show that at the L1 stage, the DD axons are completely normal (Table 1). However, by the L4 stage, defects have arisen in those same DD axons. DD axons migrate to the dorsal cord during the embryonic stage, but the animal continues to grow to an adult. It is not known how the DD axons maintain and grow with the developing body, and we wonder whether these errors can arise in the axon during this process.

How does hypergravity affect DD/VD neuron axonal development? The force of gravity itself might be strong enough to break or damage the axons. However, an acute exposure and even a long exposure to hypergravity in adult animals was not sufficient to cause damage to the axons (Fig. 2B). In addition, DD/VD motor neuron axons in *C. elegans* have the ability to repair and regenerate after damage (*Hammarlund & Jin, 2014*). In our experiments, we exposed the worms to 100G for 18 and 36 h, and then allowed the worms to recover until 60 h at 1G. This is enough time to regenerate any axons damaged by 100G force, yet we still observe hypergravity-induced axon defects. Thus, we are not certain whether high gravity induces axon damage.

We show an 18 h exposure early in development and during the 1st larval stage is sufficient to cause the axon defects (Figs. 2B and 2C). The VD neurons have yet to be born

when the critical period of hypergravity exposure occurs, yet we clearly observe defects in the VD neurons. Thus, we assume that hypergravity may be affecting another cell or tissue rather than the VD neurons themselves. Hypodermis and muscle is a major source for axon guidance cues such as UNC-6/netrin, UNC-129/TGF-beta, and LON-2/glypican (*Blanchette et al., 2015*; *Wadsworth, 2002*), and we wonder whether hypergravity can somehow be altering these cues. Further experiments to determine whether problems in axon repair, axon guidance, or growth cone formation occur as a result of hypergravity should be conducted.

In this study, we have addressed the role that high gravity conditions have on DD/VD motor neuron development. What effect does low gravity have on motor neuron development? Previous studies in rats showed that development of the dendrites of spiny motor neurons during spaceflight could be altered (*Inglis, Zuckerman & Kalb, 2000*). Similar studies to our hypergravity study can be conducted in microgravity in *C. elegans* at the genetic level. These studies could clarify whether low gravity effects on motor neuron development can also contribute to the muscle weakness and atrophy observed in astronauts during long-term spaceflight and habitation.

## ACKNOWLEDGEMENTS

Strains were provided by Jeong-Hoon Cho at Chosun University, and the Caenorhabditis Genetic Center, which is funded by NIH Office of Research Infrastructure Programs.

### Funding
This work was supported by a New Investigator Grant (2014R1A1A1005553) from the National Research Foundation of Korea (NRF) to J.I.L. Strains were provided by Jeong-Hoon Cho at Chosun University, and the CGC, which is funded by NIH Office of Research Infrastructure Programs (P40 OD010440). The funders had no role in study design, data collection and analysis, decision to publish, or preparation of the manuscript.

### Grant Disclosures
The following grant information was disclosed by the authors:
National Research Foundation of Korea (NRF) to J.I.L.: New Investigator Grant 2014R1A1A1005553.
NIH Office of Research Infrastructure Programs: P40 OD010440.

### Competing Interests
The authors declare that they have no competing interests.

### Author Contributions
- Saraswathi Subbammal Kalichamy conceived and designed the experiments, performed the experiments, analyzed the data, wrote the paper, prepared figures and/or tables, reviewed drafts of the paper.
- Tong Young Lee performed the experiments, reviewed drafts of the paper.
- Kyoung-hye Yoon conceived and designed the experiments, analyzed the data, wrote the paper, prepared figures and/or tables, reviewed drafts of the paper.
- Jin Il Lee conceived and designed the experiments, analyzed the data, wrote the paper, prepared figures and/or tables, reviewed drafts of the paper.

## Data Deposition

The raw data has been supplied as Supplemental Dataset Files.

## Supplemental Information

Supplemental information for this article can be found online at http://dx.doi.org/10.7717/peerj.2666#supplemental-information.

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
