# Peer review of "Hypergravity hinders axonal development of motor neurons in Caenorhabditis elegans"

_PeerJ, doi:10.7717/peerj.2666_

## Round 0.1 · original submission · Major Revisions

Thank you for your submission, which both reviewers have recognised as making a contribution to the field.

The reviewers have identified some key concerns with your manuscript:

Both reviewers have highlighted important issues with the data reporting and statistical analysis that need to be addressed. They also request clarification of the context of the work.

Reviewer 1 raises the possibility of including additional experiments in adult animals, and to address the mechanisms of the effects reported here. I encourage you to include the additional experiments, but if you choose not to do so, please provide a justification addressing the reviewer's concern about publishing a "self-contained" body of work.

Reviewer 1 ·

Basic reporting

The graphs (Figure 1I,J and Figure 2A,B) should be annotated to indicate statistical significance between groups using conventional line and star annotations. Statistically significant values should be included in figure legends and in text, including Figure 4.

The authors allude to an as yet unknown mechanism that results in axon defects in altered gravity (line 51), however investigation of possible mechanisms underlying this phenomenon is only mentioned in the discussion as ongoing work. The authors should consider including the proposed experiments to examine whether the susceptibility of DD/VD motor neurons is related to axon length or release of guidance cues from the surrounding tissues.

Experimental design

The authors place this work in the context of human space exploration, however do not examine how hypergravity could affect the DD/VD neurons in adult animals, with the exception of a 3hr exposure to 100G in adult worms. The inclusion of adult animals for this single experiment is not well justified. In particular, the choice of a single 3hr exposure as embryonic/larval animals did not demonstrate axonal defects following 3 hrs in hypergravity. The authors should state the rationale for using adult animals more clearly, and should consider performing additional experiments to expose adult animals to 100G for longer durations (60 and 72 hours).

Validity of the findings

There is no mention of the statistical analysis that was performed on the data in this study. This information should be provided in the materials and methods section and use of specific statistical tests should be referred to in the appropriate figure legends. The authors should not refer to ‘significant’ differences between analysed groups without providing evidence of appropriate statistical analysis.

Although Ttests are appropriate test for two groups with Gaussian distribution, this is not an appropriate test when multiple tests must be run on the same data (e.g., more than 2 groups). The authors should re-analyse the data for Figure 2A and 2B using ANOVA with an appropriate multiple comparisons test to ensure valid statistical results.

Additional comments

The editorial policies for PeerJ clearly state that submissions should be self-contained and not subdivided to increase publication count. The decision to exclude hypergravity experiments in adult animals and to not investigate potential mechanisms underlying the reported effects may be viewed as an attempt to increase publication count through the submission of multiple studies.

·

Basic reporting

The structure of the manuscript conforms to PeerJ standard and is written with mostly clear, unambiguous English language throughout. The figures are relevant, high quality and well labelled.

Perhaps the authors would consider if lines 136-140 of the text would best fit into the materials and methods section rather than the results section.

The authors have included the raw data in an Excel spread sheet, although I think it would be good to have the ‘n’ used for each experiment in the main text along with the value of the mean and standard error. This would help a reader more easily assess the research.

Critically, the authors appear not to have performed any statistical tests on the 1G versus 100G data and while I appreciate that some of the findings seem clear, these should be done as a matter of course along with an explanation as to why particular tests were chosen.

On line 119 of the text, the authors state that they ‘censured [the] all the data from these days’ when abnormally high axonal defect rates were seen in control conditions. Could the authors please clarify what they mean by ‘censured’ – was this data discarded? In addition, for the non-worm specialist, could they clarify that after these ‘occasional’ abnormal growth rates in control conditions why the backcrossing to N2 was preformed, as stated on line 120.

On the whole, the literature is relevant and well referenced, however, the authors appear to be introducing and discussing the rational for performing their experiments in the context of hypogravity (spaceflight) before abruptly switching to their own work on hypergravity. I do wonder if this link has been used to jazz up the paper, which otherwise has some solid work on axon development in hypergravity. While centrifugation has been proposed as a possible way of counteracting microgravity in long-term space flight and thus needs to be studied, this has not been made clear by the authors to a level understandable by non-gravitational experts.

I also think the authors could more appropriately set their work in the context described by Kim et al., 2007, as using the worm hypergravity paradigm to study how animals respond to mechanical stress more generally. In this context it is also interesting that 6G forces did not induce axonal defects, which suggests some robustness to mechanical stress. Something that could be of interest to people who are exposed to higher G-forces as part of their jobs, e.g. fighter pilots or formula 1 drivers.

Experimental design

This is original primary research within the scope of the journal. The research question is well defined and meaningful. As discussed in the Basic Reporting section of this review, I feel the authors could better state how this research fills an identifiable knowledge gap.

The work as far as I can tell has been performed to a high technical and ethical standard and the methods have been described with sufficient detail and information to replicate.

Validity of the findings

The data has 1G controls, but not statistical tests have been performed. This should be corrected before publication in the journal.

The conclusions are well stated, although for clarity and grammatical correctness, I would remove the sentence starting on line 305, “However, no evidence can account for these explanations” and add, “We speculate that one possibility…” into the sentence beginning on line 301, to clearly identify this as speculation.

Additional comments

Due to the importance of including the correct statistical tests I have recommended 'major revisions'. The research appears to be well conducted and solid, so the only other thing I would recommend revising in addition to the statistical tests is modifying the context that the work is set, to be far less linked to spaceflight (although this link should not be lost) and instead emphasise what appears to be the more relevant mechanical stress/hypergravity context.

---

## Round 0.2 · accepted · Accept

Congratulations on acceptance of your manuscript and thank you for thoroughly addressing the reviewers' concerns.

Reviewer 1 ·

Basic reporting

The authors have improved the manuscript through the revision process and now present a clear and logical set of experiments to understand how hypergravity affects the developing C.elegans motor system. The revised changes to statistical reporting and graph annotation now clearly highlight the significance of the findings.

Experimental design

The authors have addressed the concerns relating to the exclusion of adult animals for this study and have now clearly justified the use of developing C.elegans.

Validity of the findings

Revised changes to statistical analysis and inclusion of additional experiments strengthen the study that now presents a thorough and comprehensive report of the effects of hypergravity on DD/VD motor neurons..

Additional comments

The authors have improved the manuscript and have addressed the concerns of the initial revision. The revised manuscript presents a clear and logical flow of experiments. The decision to exclude specific time-points or ages are now well-justified. The revisions to the discussion section draw a distinct boundary around the work that has been performed for this publication and those experiments that should be considered for future study.

·

Basic reporting

I believe that the revised version of this manuscript meets all of PeerJ's basic reporting criteria.

Experimental design

It appears that the revised manuscript meets PeerJ's standards for experimental design

Validity of the findings

Statistical tests have been added in the revised manuscript. I believe that the manuscript now meets PeerJ's standards in this area.

Additional comments

Thank you for addressing the reviewers comments. I feel that the changes you have made to the manuscript makes it a stronger piece of work and easier to understand for a non-specialist.